# STAR ATTENTION: EFFICIENT LLM INFERENCE OVER LONG SEQUENCES

## ABSTRACT

Inference with Transformer-based Large Language Models (LLMs) on long sequences is both costly and slow due to the quadratic complexity of the self-attention mechanism. We introduce Star Attention, a two-phase block-sparse approximation that improves computational efficiency by sharding attention across multiple hosts while minimizing communication overhead. In the first phase, the context is processed using blockwise-local attention across hosts, in parallel. In the second phase, query and response tokens attend to all prior cached tokens through sequence-global attention. Star Attention integrates seamlessly with most Transformer-based LLMs trained with global attention, reducing memory requirements and inference time by up to 11x while preserving 95-100% of accuracy.

## 1 INTRODUCTION

Recent Large Language Models (LLMs) can support contexts up to millions of tokens in length (Gemini-Team, 2024; Anthropic, 2024; Meta-AI, 2024), unlocking applications such as repository-level code analysis, multi-document summarization, and large corpus retrieval. However, processing such long sequences with LLMs requires substantial computational and memory resources due to the quadratic complexity of the self-attention mechanism.

To address these challenges, various techniques have been proposed to reduce memory usage and increase inference speed. For example, *Flash Attention* introduces an efficient GPU block-wise implementation of the global attention, achieving significant reductions in memory overhead and runtime (Dao et al., 2022; Dao, 2024). *Ring Attention* further extends this idea by distributing the computation of self-attention and feed-forward modules across multiple devices, cleverly overlapping communication with shard-local attention computations to enhance scalability (Liu et al., 2024a; Beltagy et al., 2020). More broadly, distributed strategies such as tensor, pipeline, sequence, and data parallelism have been proposed to divide compute effectively across multiple machines (Shoeybi et al., 2019; Huang et al., 2019; Li et al., 2023; Meta-AI, 2021).

Several prior works have shown that the attention matrix can be approximated with sparse attention mechanisms reducing the algorithmic complexity from quadratic to linear or log-linear. Child et al. (2019) significantly reduces the complexity of attention by leveraging sparse factorizations and (Choromanski et al., 2021) approximates attention using kernel-based methods. (Beltagy et al., 2020) employs sliding window attention and global tokens for efficient long-sequence processing while Xiao et al. (2024) adapts it for real-time long-sequence generation utilizing attention sinks. Complementing these approaches, memory-efficient techniques have also emerged. Key-value (KV) cache compression (Dai et al., 2019; Ge et al., 2024; Munkhdalai et al., 2024; Sun et al., 2024; Liu et al., 2024b) and low-rank approximations (Srebro & Jaakkola, 2003) trade precision for reduced memory usage.

We introduce Star Attention, a novel algorithm for efficient LLM long-context inference [1]. This method is based on the observation that LLM inference usually has two stages: (1) prompt encoding, where the model processes input and stores KV vectors in the cache and (2) token generation, where model attends to the KV cache and autoregressively generates new tokens while updating the cache with the new KV vectors. In many long-context tasks, the input consists of a long context followed by a short query and a short answer. The information needed for answering the query is often

---

[1]Code will be open-sourced

localized within small parts of the context, meaning context tokens need only attend to nearby tokens, while query tokens need to attend to all prior tokens. Based on this observation, *Star Attention* utilizes a two-phase approach shown in Figure 1:

1. **Context Encoding**: The context is divided into contiguous blocks and distributed across "context" hosts, with each host also receiving a copy of the first block (an "*anchor block*"). Hosts compute self-attention only for their assigned blocks, without communicating with each other, reducing attention complexity from quadratic to linear with respect to context length. This distributed processing is similar to Ring Attention (Liu et al., 2024a) but without the "ring" communication during context encoding (Figure 1a).

2. **Query Encoding and Token Generation**: The query is replicated across all hosts where it initially attends to the KV cache on each host. Global attention is then computed by aggregating the results at a designated "query" host by efficiently communicating a single vector and scalar per token from each context host. Only the query host updates its KV cache during this stage (Figure 1b).

Star Attention enables the context length to scale linearly with the number of hosts by distributing the context processing across multiple hosts. Star Attention is compatible with most Transformer-based LLMs trained with global attention, operating seamlessly out-of-the-box without additional model fine-tuning. We evaluate Star Attention for Llama3.1-8B and Llama3.1-70B (Meta-AI, 2024) on several long-context benchmarks. Star Attention achieves up to 11 times faster inference while maintaining 95-100% of the baseline accuracy. Furthermore, Star Attention can be combined with other LLM optimization methods like Flash Attention or KV cache compression, allowing for additional speedup enhancements during inference.

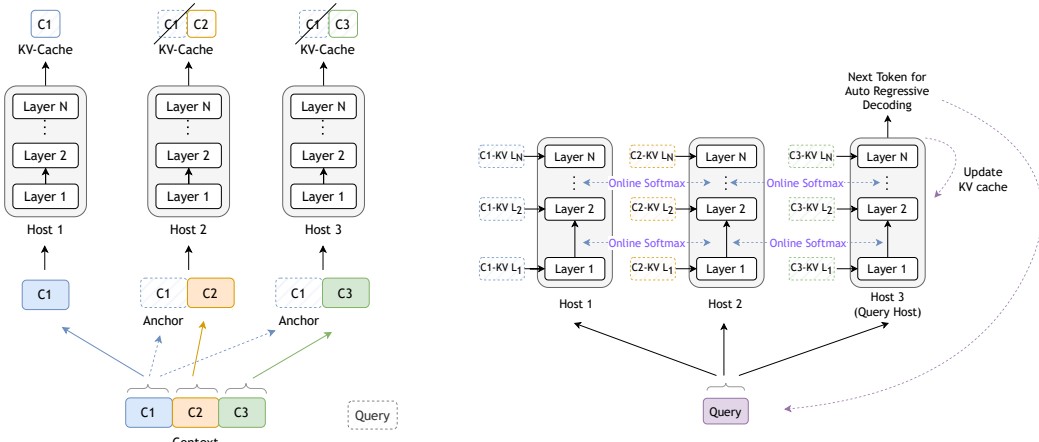

(a) **Phase 1**: Local Context Encoding with Anchor Blocks

(b) **Phase 2**: Query Encoding and Output Generation with Global Attention

Figure 1: Star Attention inference flow. All devices in the system are grouped into hosts where one of the hosts is labeled as the "query" host. The input sequence is processed in two phases. *Phase 1 - context encoding*. The context portion of the input is partitioned into smaller blocks and distributed across hosts. All blocks, except the first, are prefixed with the initial block, called the "anchor" block. Each host processes its assigned block and stores the non-anchor portion of the KV cache. *Phase 2 - query encoding and token generation*. The input query is broadcast to all the hosts, where in each host, it first attends to the local KV cache computed during phase one. Then the "query" host computes global attention by aggregating the softmax normalization statistics from all the hosts. This process is repeated for each generated token.

## 2 STAR ATTENTION ALGORITHM

Star Attention operates in two phases: (1) *Context Encoding*, where the long context is divided into contiguous blocks and is processed with local blockwise attention, and (2) *Query Encoding*

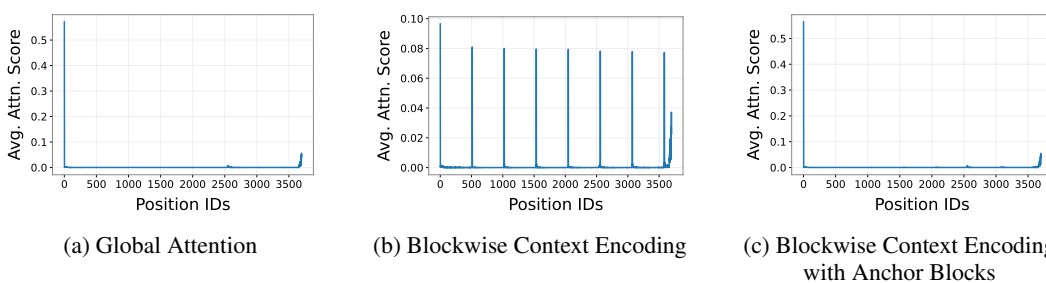

Figure 2: Block sparsity pattern for a sequence partitioned into 5 context blocks $c_i$ and a query block $q$. Each context block attends only to itself and the "anchor block" whereas the query attends to the entire input.

| (a) Global Attention | (b) Blockwise Context Encoding | (c) Blockwise Context Encoding with Anchor Blocks |
|---|---|---|

Figure 3: Attention distribution along the sequence length for context encoded with different strategies in phase 1 of Star Attention. (a) Global attention shows a spike at the start, corresponding to the attention sink. (b) Star Attention without anchor blocks shows several attention sinks present at the beginning of each block. (c) Star Attention with anchor blocks shifts sinks to anchor tokens, resulting in an attention distribution approximating global attention. In the plot, the input sequence (4K tokens) is divided into 512-token chunks.

*and Token Generation*, where the query is processed, and answer tokens are generated using global attention. Below, we detail each phase of the algorithm.

## 2.1 PHASE 1: CONTEXT ENCODING

Given an input sequence comprising a context $c$ followed by a query $q$, the context $c$ is divided into $n$ contiguous blocks: $c = [c_1, c_2, \ldots, c_n]$, where each block $c_i$ contains $b$ tokens. We introduce an *anchor block* mechanism, in which, each block—except the first—is prefixed with the first block $c_1$ of the sequence, referred to as the anchor block. This concatenation forms an augmented context $c'$:

$$c' = [c_1, (c_1\ c_2), (c_1\ c_3), \ldots, (c_1\ c_n)]$$

where each augmented block $c_i'$ contains $2b$ tokens: $b$ tokens from the anchor block $c_1$ followed by $b$ tokens from the current block $c_i$ (Figure 2). The positional indices of $c_1$ are preserved, ensuring that its tokens retain their original position indices $[0, 1, \ldots, b-1]$. The augmented blocks are distributed across compute hosts, where each host computes attention over the $2b$ tokens from its assigned block $c_i'$ and generates the corresponding key-value (KV) vectors. While KVs for the anchor block $c_1$ are discarded, the KVs for the current block $c_i$ are retained in the cache.

We observe that, without anchor blocks—i.e., applying blockwise attention only to the original context $c$—the model fails to generate correct outputs. We conjecture this failure is due to the incorrect approximation to the attention patterns observed during phase 2 (Figure 3b), where multiple attention spikes, known as attention sinks (Xiao et al., 2024), are distributed across the sequence. These spikes occur because each block is processed independently, creating an attention sink at the start of each block. As a result, the model struggles to effectively focus on relevant parts of the context. To address this issue, we prefix the blocks with the anchor block $c_1$, shifting the attention sinks to the anchor tokens. By discarding the KVs of the anchor tokens the intermediate attention sinks are removed ensuring the attention distribution of block-local attention (Figure 3c) closely approximates global attention (Figure 3a) while maintaining the computational efficiency of blockwise processing.

## 2.2 Phase 2: Query Encoding and Token Generation

In phase 2, global attention is employed to encode the query and generate output tokens by using a distributed softmax algorithm that eliminates the need to transfer KV cache between hosts (Figure 1b). A designated query-host $h_q$ coordinates this computation. The query is broadcast to all hosts and transformed into the sequence $Q \in \mathbb{R}^{l_q \times d}$, where $l_q$ is the query length, and $d$ is the attention head dimension. Each host $h$ computes the local attention output $A_h$ for the query $Q$ using its local key-value pairs $K_h, V_h \in \mathbb{R}^{l_k \times d}$, where $l_k$ is the sequence length of the KV cache. The local attention is computed as:

$$A_h = \left( \frac{\exp\left(\frac{QK_h^\top}{\sqrt{d}}\right)}{\sum_{k=1}^{l_k} \exp\left(\frac{QK_{h,k}^\top}{\sqrt{d}}\right)} \right) V_h \tag{1}$$

In addition to $A_h$, each host also stores the sum of the exponents $s_h$ from the the local softmax operation (the denominator from Equation 1):

$$s_h = \sum_{k=1}^{l_k} \exp\left(\frac{QK_{h,k}^\top}{\sqrt{d}}\right) \tag{2}$$

The query-host $h_q$ gathers the local attention $A_h$ and the sums of exponents $s_h$ from all hosts:

$$A = [A_1, A_2, \ldots, A_H]$$

$$s = [s_1, s_2, \ldots, s_H]$$

The global softmax denominator, $s_{\text{global}}$, is then computed as the sum of all local exponents:

$$s_{\text{global}} = \sum_{h=1}^{H} s_h \tag{3}$$

The query-host uses $s_{\text{global}}$ to aggregate the local attentions to compute the global attention:

$$A_{\text{global}} = \sum_{h=1}^{H} \frac{s_h}{s_{\text{global}}} A_h \tag{4}$$

This method ensures that the global attention scores are normalized correctly across all hosts. It requires the communication of only a single scalar $s_h$ (the local sum of exponents) and a vector $A_h$ (the local attention) per token. In practice, the *log-sum-exp* method from online softmax (Milakov & Gimelshein, 2018) can be used to maintain the numerical stability during global attention aggregation. This distributed approach enables efficient computation by minimal data transfers between hosts.

**Output generation and cache update.** After computing the global attention output, the query-host $h_q$ generates the next token and its KV cache is updated with the key and value vectors of the new token. This process is repeated for each generated token.

This two-phase mechanism—local context encoding with anchor blocks in Phase 1 followed by global query encoding with token generation in Phase 2—gives significant improvements in inference speed, while keeping the accuracy close to the global attention.

## 3 Experiments

We evaluate Star Attention on several Llama-based models with sequence lengths ranging from 16K to 1M tokens on RULER (Hsieh et al., 2024) and BABILong (Kuratov et al., 2024) benchmarks. We begin by comparing accuracy and the speed achieved by Star Attention versus baseline - Ring attention. Further, we investigate the impact of varying block sizes on accuracy, illustrating the trade-off between accuracy and speedup. Finally, we conduct a detailed analysis of challenging and favorable cases for Star Attention by examining distinct RULER task categories.

| Model | Seq. Len. (K) | Block Size (K) | Ring-Attn Acc.(%) | Star-Attn Δ Acc. | Star-Attn Δ Speedup |
|---|---|---|---|---|---|
| Llama-3-8B-Instruct, 1048K Gradient.ai (2024) | 16 | 4 | 86.12 | +2.47% | 1.1x |
| | 32 | 8 | 82.52 | +1.54% | 1.2x |
| | 64 | 16 | 79.05 | +1.28% | 1.8x |
| | 128 | 32 | 77.39 | +1.23% | 2.7x |
| Llama-3.1-70B-Instruct, 128K Meta-AI (2024) | 16 | 4 | 95.09 | -2.85% | 1.7x |
| | 32 | 8 | 94.61 | -2.70% | 2.0x |
| | 64 | 16 | 88.54 | -1.63% | 4.7x |

Table 1: Star Attention vs Ring Attention (baseline) accuracy and relative inference speed-up. The Δ for Star Attention shows the relative accuracy improvement (+) or degradation (-). We set block size to one-quarter of the sequence length. Star Attention achieves significant speedup over Ring Attention while maintaining the accuracy. For larger models, the speedup of Star Attention is even more pronounced.

### 3.1 Setup

**Models.** We benchmark the base and instruct variants of the Llama-3.1 8B model which support context lengths up to 128K tokens (Meta-AI, 2024). In addition, we evaluate two Gradient.ai models that extend Llama-3-8B's context up to 1M tokens Gradient.ai (2024). To access the scalability of our method, we also evaluate the Llama-3.1-70B-Instruct model. We observe that large LMs achieve even greater speedups with Star Attention on long context tasks.

**Baseline.** We compare Star Attention with Ring Attention (Liu et al., 2024a). Ring Attention computes global block-wise attention by having each host communicate its respective KV cache in a ring pattern across all the hosts . More details regarding our baseline selection in Appendix C.1.

**Configuration.** We implement Star Attention using HuggingFace Transformers library (Wolf et al., 2020). All experiments are done on A100 GPUs with bfloat16 precision. More details on the experiment configuration are in Appendix C.

**Evaluation Benchmarks.** We use the RULER benchmark for evaluation. It consists of 13 tasks categorized into 4 domains: Needle-in-a-Haystack (Retrieval), Multi-Hop Tracing, Aggregation, and Question Answering (Hsieh et al., 2024). Additionally, we report results on the BABILong benchmark, which encompass tasks where multiple supporting facts encoded in the context are required to generate accurate answers (Kuratov et al., 2024). Further details on the benchmarks and specific tasks can be found in Appendix B.

### 3.2 Results

Table 1 provides relative speedup and accuracy achieved by Star Attention versus Global (Ring) Attention from 16K to 128K tokens on the RULER benchmark. In each setting, the context block size and anchor block size are set to one-quarter of the total sequence length. Star Attention achieves similar accuracy to full global attention, with relative accuracy degradation limited to 0-3% while also giving upto 5x inference speedup. This demonstrates that Star Attention effectively preserves the model's ability to retrieve relevant information, even with a significantly reduced context window. In case of larger models, such as Llama-3.1-70B Instruct, we find that these models achieves even greater speedups at any given sequence length while maintaining similar levels of accuracy degradation. We discuss Star Attention's strengths and limitations based on RULER subtasks in Section 3.4. Full RULER scores for all models can be found in Appendix E.

Extending this analysis to other benchmarks and models, we evaluate Star Attention on the BABILong benchmark as well using Llama-3.1-8B-Instruct, Llama-3.1-8B-Base, and the gradientai-Llama-3-8B-Instruct-262K model. We have a similar observation here that Star Attention achieves similar accuracy to full global attention, with accuracy degradation limited to 0-3% across all tasks up to 128K tokens, as shown in Figure 4.

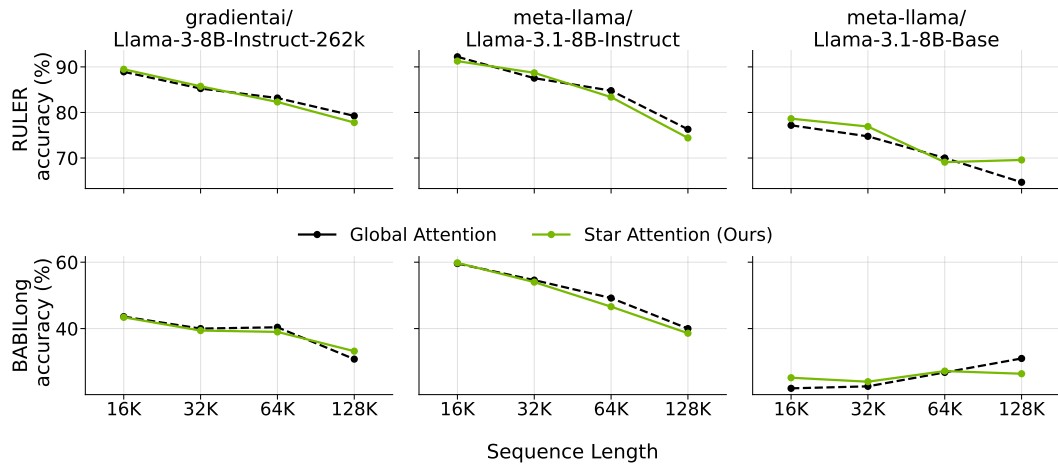

Figure 4: Accuracy (%) of Star Attention on RULER and BABILong evaluated on sequence lengths of 16K, 32K, 64K, and 128K. In all experiments, the block size and anchor block size are set to one-quarter of the total sequence length. Results using the Llama-3-8B-Instruct-262k, Llama-3.1-8B-Instruct and Llama-3.1-8B-Base models demonstrate that Star Attention retains 95-100% of the accuracy of global attention, and in some cases, even outperform it.

However, we observe several anomalies with the Llama-3.1 8B base model on the BABILong benchmark. There is a significant improvement at the 16K sequence length, but a severe drop at 128K. These fluctuations likely stem from the benchmark's format-specific requirements for generating answers, which pose challenges for base models since they are not optimized for instruction-following tasks. As a result, scores for base models, especially at longer sequence lengths, may be less reliable. More details provided in Appendix D.

## 3.3 TRADE-OFF BETWEEN ACCURACY AND SPEED

Figure 5, illustrates the effect of varying block sizes during context encoding with sequence length of 128K tokens. Larger block sizes correlate to higher accuracy with Star Attention. This trend is consistent across all sequence lengths in our experiments.

From our experiments, we observe that setting the block size to approximately one-quarter of the total sequence length strikes an optimal trade-off between accuracy and speed. However, for sequence lengths exceeding 128K, as shown in Table 2, we fix the block size at 32K tokens to prioritize speedup, allowing for some acceptable accuracy degradation. Similarly, for larger models such as the Llama-3.1-70B-Instruct, we limit the block size to 16K tokens. The choice of block size is dependent on the user on how much accuracy can be traded for improved speed. As the block size increases, Star Attention's performance approaches that of full global attention, providing users with flexibility in balancing computational efficiency with accuracy. Additional details regarding the experimental setup are provided in Appendix C.2.

## 3.4 IN-DEPTH ANALYSIS ON RULER TASK CATEGORIES

In this section we investigate the strengths and limitations of Star Attention, using different categories of tasks within RULER. The benchmark has five primary categories: Single-NIAH, Multi-NIAH, Multi-Hop Tracing, Aggregation, and Question Answering (QA). Figure 6 presents categorical results of RULER for the Llama-3.1-8B-Instruct model on sequence length of 32K. The trend is consistent across all sequence lengths (16K, 32K, 64K, and 128K), as detailed in Appendix E (Figure 8). Notably, Star Attention achieves scores nearly identical to global attention in Single-NIAH tasks. However, in more complex tasks such as Multi-NIAH and QA, it shows slight decline in performance, with reductions ranging from 1.6% to 4.9% in Multi-NIAH and 0.9% to 6.8% in QA tasks. Despite these challenges, Star Attention consistently retains overall 95-100% accuracy of global attention.

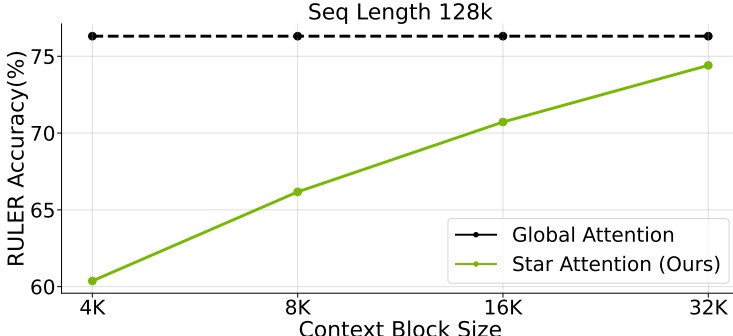

Figure 5: Effect of block size on the accuracy of Star Attention on the RULER benchmark with block sizes ranging from 4K to 32K tokens for Llama-3.1-8B instruct model at sequence length of 128K. In each setting, the anchor block size matches the context block size. The results indicate that larger context block sizes are positively correlated with improved accuracy.

| Model | Seq. Len. (K) | Block Size (K) | Ring-Attn Acc. (%) | Star-Attn Δ Acc. | Δ Speedup |
|---|---|---|---|---|---|
| Llama3-8B-Instruct, 1048K Gradient.ai (2024) | 128 | 32 | 77.39 | +1.23% | 2.7x |
| | 256 | 32 | 74.44 | -1.04% | 10.8x |
| | 512 | 32 | 69.30 | -9.71% | 16.2x |
| | 1024 | 32 | 63.70 | -8.36% | 16.9x |
| Llama-3.1-70B-Instruct, 128K Meta-AI (2024) | 64 | 16 | 88.54 | -1.63% | 4.7x |
| | 128 | 16 | 65.29 | -11.44% | 8.7x |

Table 2: Accuracy versus speed trade-off for Star Attention compared to Ring (Global) Attention on RULER. The Δ for star attention shows the relative accuracy degradation and the relative speedup compared to global attention. When the block size remains fixed and the as sequence length increases, Star Attention achieves exponential speedup over Ring (Global) Attention at the cost of slightly more accuracy degradation. It is upto the user to decided how much accuracy they want to trade-off for speed by setting the block size.

Tasks such as Multi-Hop Tracing and Aggregation necessitate an in-depth comprehension of context. Multi-Hop Tracing poses a significant challenge for Star Attention, as it requires the model to propagate information across multiple hops within the sequence, demanding effective inter-block communication. However, Star Attention lacks inter-block communication, relying solely on global attention between the query and segregated KV caches within each block. Due to this, the performance degradation is considerable compared to global attention.

Aggregation tasks, encompassing Common and Frequent Words Extraction, assess models' ability to aggregate relevant information within long-range contexts. Star Attention yields significant performance improvements across all sequence lengths. This enhancement stems from Star Attention's chunk-based context division, enabling local attention within each chunk to strengthen summarization capabilities. Effective chunk-wise summarization in Phase 1 facilitates global attention's information aggregation in Phase 2.

## 4 ABLATION STUDY

The ablation experiments focus on the Needle-in-a-Haystack (NIAH) task, which tests a model's ability to answer queries based on a small, relevant piece of information ("needle") embedded within a large context ("haystack"). To increase the task's complexity, we explore three variations from the RULER benchmark (Hsieh et al., 2024): Single-NIAH, Multi-key NIAH, and Multi-query NIAH.

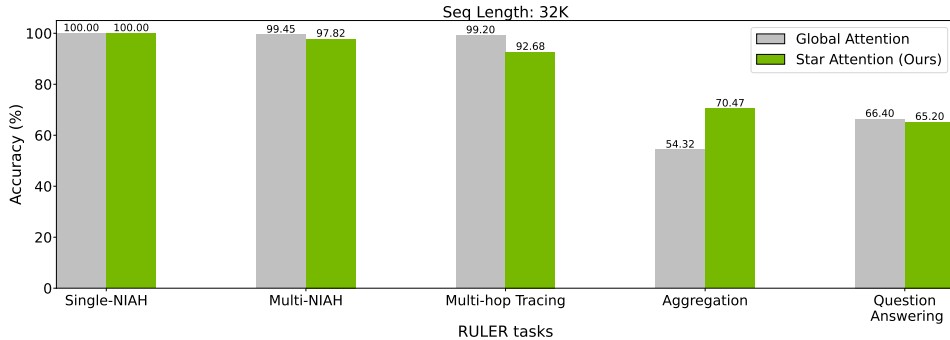

Figure 6: Accuracy (%) of Star Attention using the Llama-3.1-8B-Instruct model on the 5 categories of tasks in RULER on sequence lengths of 32K. In all experiments, the block size and anchor block size are set to one-quarter of the total sequence length. For the NIAH and QA tasks, Star Attention retains upto 95-100% accuracy of the baseline. The Multi-Hop Tracing task becomes quite challenging since it requires inter-block communication. In aggregation tasks, Star Attention show significant improvement as distributed local attention helps the model in such summarization tasks. The trend is consistent to other sequence lengths as shown in Appendix E (Figure 8)

### 4.1 POSITION AND CONTENT OF ANCHOR BLOCK

In this section, we explore the role of anchor blocks during Phase 1 that enables Star Attention to approximate global attention behavior. As outlined in Section 2.1, anchor blocks are crucial in managing the attention spikes generated at the start of each context block, helping Star Attention approximate global attention (see Table 3 ) Drawing from the hypotheses on sink tokens in Xiao et al. (2024), we consider two potential explanations for the effectiveness of anchor blocks: (1) the model may develop a bias toward the absolute position of the anchor block, or (2) the semantic content of the anchor block is essential for maintaining performance. To better understand how anchor blocks enable Star Attention to approximate global attention distribution, we test both the hypotheses. We conduct experiments on the Llama-3.1-8B-Instruct model, varying both the position and content of the anchor block. We evaluate two configurations: a block size of 16K for sequences of length 64K, and a block size of 32K for sequences of length 128K, in both the cases, with anchor block size matching the context block size.

**Position of anchor block**: Here, we fix the content of the anchor block to the first context block and vary its position IDs. We test three scenarios : (1) the position IDs are randomly sampled from the range [0, starting position of the current block] (e.g., for a block starting at position 32K, position IDs are sampled from [0, 32K] ); (2) the position IDs are derived from the previous block (e.g., for a block of size 16K starting at position 32K, position IDs are sampled from [16K, 32K] ); (3) the position IDs are fixed to the first block (our proposed approach). As shown in Table 3, varying the position of the anchor block has minimal impact on accuracy.

**Content of anchor block**: We fix the position IDs of the anchor block to that of the first block but vary its content. We explore several configurations (as shown in Table 3): (i) a single repeated token (e.g., ` `, ` the`, or `.`); (ii) random tokens; (iii) shuffling the tokens of the first block; and (iv) using the original first block content (the proposed approach). Our results show that the content of the anchor block significantly impacts performance, with the original first block content yielding the best results. This outcome suggests that since global attention is performed during Phase 2, it is important for the local context blocks to attend to anchor blocks whose content reflects what the model would see during global attention.

**Previous block as anchor block**: To examine the roles of both position and content, we experiment with using the previous block as the anchor block. For example, for a block of size 16K starting at position 32K, the anchor block would be the block with position IDs from 16K to 32K. This configuration has lower accuracy comparing to using the first block as the anchor(Table 3).

In summary, we found that while the positional placement of the anchor block is not important , its content is critical for optimal performance.

| Experiments | RULER-NIAH (%) | | | |
| --- | --- | --- | --- | --- |
| | 64K | Δ64k | 128k | Δ128k |
| Global attention | 99.50 | - | 98.49 | - |
| No anchor block | 60.11 | -39.59% | 73.75 | -25.12% |
| Content set to first-block, position IDs are: | | | | |
|     randomly sampled from [0, current_block) | 96.79 | -2.72% | 97.16 | -1.35% |
|     same as previous block | 97.35 | -2.16% | 96.80 | -1.71% |
|     **same as first block** | 97.61 | -1.90% | 97.54 | -0.96% |
| Position IDs set to first-block, content is: | | | | |
|     constant token (ex: ' ' or ' the' or '.' ) | 0.00 | -100.00% | 0 | -100.00% |
|     random tokens | 90.55 | -8.99% | 82.63 | -10.15% |
|     shuffled first block tokens | 92.96 | -6.57% | 90.76 | -3.26% |
|     **first block tokens** | 97.61 | -1.90% | 94.94 | -0.96% |
| Previous-block used as anchor | 94.20 | -5.33% | 96.13 | -2.40% |

Table 3: Experiments on analyzing the impact of varying the position and content of the anchor block with the LLaMA-3.1-8B-Instruct model, with a block size of 16K for 64K sequence length, and 32K for 128K sequence lengths. In each setting, the size of the anchor block matches the context block size. The Δ for star attention shows the relative accuracy degradation compared to global attention. The experiments are categorized into 4 groups: (i) absence of anchor block; (ii) varying the position IDs; (iii) varying the content; (iv) varying both the position and the content. Results indicate that while the anchor block's position is not critical, its content is essential for optimal performance.

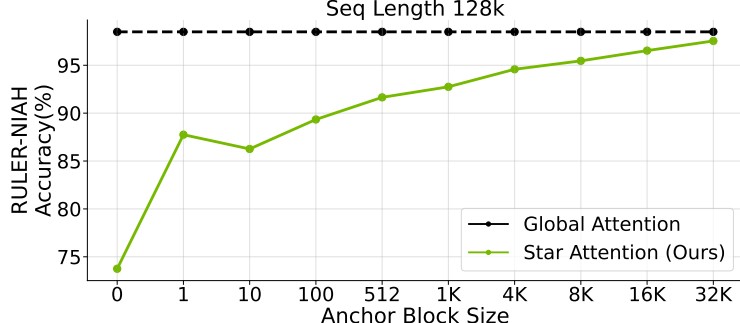

Figure 7: Effect of anchor block size on the accuracy of Star Attention on the RULER-NIAH benchmark with the Llama-3.1 8B Instruct model. In these experiments, the context block size is fixed to 32K for sequence length 128K, respectively. Results indicate that larger anchor block sizes lead to improved accuracy. The observed trend holds for all sequence lengths in our experiments.

## 4.2 SIZE OF ANCHOR BLOCK

As discussed in Section 3.3, larger block sizes improve the accuracy of Star Attention. In this section, we analyze the impact of varying anchor block size while maintaining a fixed block size of 32K for a sequence length of 128K. As illustrated in Figure 7, increasing the anchor block size enhances model accuracy, with the best performance observed when the anchor block size equals the context block size. Although Figure 3b demonstrates that attention spikes predominantly occur in the first few tokens, reducing the number of tokens in the anchor block leads to a substantial drop in performance. This suggests that a larger anchor block is critical for maintaining model accuracy, despite attention spikes being concentrated at the beginning of the sequence. This observation implies that the anchor block's effectiveness is not solely due to its role in managing attention sinks but may involve other underlying factors. These findings remain consistent across both base and instruct models, as well as for all sequence lengths. Further investigation into why the anchor block size must be equivalent to the context block size is left for future work.

## 5 CONCLUSION

In this paper, we introduced Star Attention, a novel block-sparse attention mechanism designed to enable efficient inference on long sequences in transformer-based LLMs. The method operates in two phases: (1) context tokens are processed using blockwise-local attention, with the context segmented into blocks where each block is prefixed with an anchor block; and (2) then the query and response tokens attend to all prior cached tokens through sequence-global attention. Star Attention delivers up to 11x speedup over Ring Attention while maintaining 95-100% accuracy, significantly enhancing both memory efficiency and inference speed. Scaling Star Attention to longer sequences (up to 1M) and larger models, we observe even greater speedups while preserving similar levels of accuracy. Despite these advances, several open questions remain. The role and optimal size of anchor blocks relative to context blocks require further exploration. Additionally, while Star Attention performs effectively with block sizes set to one-quarter of the sequence length, accuracy degrades when using smaller blocks on longer sequences. Future work will focus on refining the anchor block mechanism and improving performance on more complex long-context tasks to enhance the scalability and robustness of Star Attention.

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

# A STAR ATTENTION PSEUDO-CODE

---

**Algorithm 1** Star Attention - Phase 1: Context Encoding

---

**Require:** Context $c$, Block size $b$
1: $L \leftarrow \text{length}(c)$
2: Split $c$ into $n = \lceil L/b \rceil$ blocks, such that $c = [c_1, c_2, \ldots, c_n]$     ▷ Each block has upto $b$ tokens
3: **for** $i = 2$ to $n$ **do**
4:     $c_i' \leftarrow (c_1, c_i)$                  ▷ Each block $c_i$ is prefixed with anchor block $c_1$
5: **end for**
6: **for** each host concurrently **do**
7:     Initialize an empty list $kv$
8: **end for**
9: Distribute augmented blocks $[c_1', c_2', \ldots, c_n']$ across all hosts
10: **for** each host concurrently **do**               ▷ Parallel processing on each host
11:     **for** each assigned block $c_i'$ **do**
12:        Compute attention over $2b$ tokens in $c_i'$
13:        Generate KV cache for $c_i'$
14:        Discard KV cache for anchor block $c_1$
15:        Append remaining KV cache (for $c_i$) to $kv$
16:     **end for**
17: **end for**

---

**Algorithm 2** Star Attention - Phase 2: Query Encoding and Token Generation

---

**Require:** Query tokens $q$, number of output tokens $n_o$, KV cache $kv_h$ of each host from Phase 1
1: Designate one host as the query-host $h_q$
2: Broadcast query tokens $q$ to all hosts
3: Initialize $input\_tokens \leftarrow q$
4: Initialize $output\_tokens \leftarrow []$
5: **for** $i = 1$ to $n_o$ **do**                 ▷ Generate $n_o$ output tokens
6:     **for** each transformer layer **do**         ▷ Process through all transformer layers
7:        **for** each host $h$ concurrently **do**     ▷ Execute parallel computations on each host
8:           Compute query, key, and value vectors $(Q, K, V)$ using $input\_tokens$
9:           **if** $h = h_q$ **then**               ▷ If this is the query-host
10:             Append the new $K$ and $V$ vectors to $kv_{h_q}$
11:           **end if**
12:           Compute local attention scores $A_h$ for query $Q$ using the local KV cache $kv_h$
13:           Compute local log-sum-exp $s_h$ (logarithm of the softmax denominator)
14:        **end for**
15:        Gather all $A_h$ and $s_h$ from hosts: $s = [s_1, s_2, \ldots, s_H], \quad A = [A_1, A_2, \ldots, A_H]$
16:        Initialize $s_{\text{global}} \leftarrow s_1, A_{\text{global}} \leftarrow A_1$
17:        **for** $h = 2$ to $H$ **do**            ▷ Aggregate attention scores across hosts
18:           Update global log-sum-exp $s_{\text{global}}$ using online softmax:
$$s_{\text{global}} \leftarrow s_{\text{global}} + \log\left(1 + \exp(s_h - s_{\text{global}})\right)$$
19:           Update global attention scores:
$$A_{\text{global}} \leftarrow \exp(s_h - s_{\text{global}}) \cdot A_{\text{global}} + \exp(A_h - s_{\text{global}}) \cdot A_h$$
20:        **end for**
21:     **end for**
22:     Generate the next output token from the last transformer layer
23:     Append the new output token to $output\_tokens$
24:     Set $input\_tokens \leftarrow [\text{new output token}]$
25: **end for**
26: **return** $output\_tokens$

---

## B    EVALUATION BENCHMARKS

We evaluate Star Attention on the RULER benchmark which comprises 13 tasks covering domains such as Needle-in-a-Haystack (Retrieval), Multi-Hop Tracing, Aggregation, and Question Answering. Each task comprises 500 samples. For the ablations, we choose four Needle-In-A-Haystack (NIAH) tasks where Paul Graham essays serve as the distractor text (haystack): Single 2, Single 3, MultiKey 1, and MultiQuery. In these tasks, a key-value pair is concealed within a long context, and the model must identify the value corresponding to the key based on the provided input query. Table 4 presents the configurations of all the tasks in RULER.

| Category | Task Name | Haystack Type | Keys Type | # | Values Type | # | # Outputs |
|---|---|---|---|---|---|---|---|
| NIAH (Retrieval) | Single 1 | noise | words | 1 | numbers | 1 | 1 |
| | Single 2 | book | words | 1 | numbers | 1 | 1 |
| | Single 3 | book | words | 1 | uuids | 1 | 1 |
| | MultiKey 1 | book | words | 4 | numbers | 1 | 1 |
| | MultiKey 2 | line | words | $\infty$ | numbers | 1 | 1 |
| | MultiKey 3 | kv | uuids | $\infty$ | uuids | 1 | 1 |
| | MultiValue | book | words | 1 | numbers | 4 | 1 |
| | MultiQuery | book | words | 4 | numbers | 1 | 4 |
| Multi-Hop Tracing | Variable Tracking | | | | – | | |
| Aggregation | Common Words Extraction | | | | – | | |
| | Frequent Words Extraction | | | | – | | |
| Question Answering | QA 1 (squad) | | | | – | | |
| | QA 2 (hotpotqa) | | | | – | | |

Table 4: Configuration of RULER tasks

Additionally, we also evaluate our method on the BABILong benchmark. In BABILong, we choose 5 tasks (shown in Table 5), each containing a 1000 samples. These tasks are generated by simulating a set of characters and objects engaged in various movements and interactions across multiple locations. Each interaction is represented by a factual statement, and the objective is to answer questions based on the facts derived from the current simulation.

| Task | Name | # Facts per task |
|---|---|---|
| qa1 | single supporting fact | 2 - 10 |
| qa2 | two supporting facts | 2 - 68 |
| qa3 | three supporting facts | 4 - 32 |
| qa4 | two arg relations | 2 |
| qa5 | three arg relations | 2 - 126 |

Table 5: Configuration of tasks in BABILong

## C    EXPERIMENT DETAILS

### C.1    BASELINE COMPARISON

Our implementation utilizes the HuggingFace Transformers library (Wolf et al., 2020), which currently lacks support for multi-node inference. As a result, when performing inference with the Llama-3.1 8B model using standard causal autoregressive generation on sequences exceeding 64K tokens with bfloat16 precision across 8 A100 GPUs, we encounter out-of-memory (OOM) errors. Given these limitations, we adopt Ring Attention as a practical and relevant baseline for evaluating Star Attention's performance on sequences up to 1 million tokens in length.

Table 6 presents the time per sample for vanilla autoregressive generation, Ring Attention, and Star Attention across sequence lengths ranging from 16K to 128K. The results indicate that both Ring and Star Attention can process sequences up to 128K tokens on 8 A100 GPUs, whereas vanilla autoregressive inference encounters OOM issues beyond 64K tokens. For sequence lengths below 32K, vanilla inference is faster than the distributed attention mechanisms, primarily due to the GPU communication overhead incurred in the distributed setups. However, in long context scenarios i.e. on sequence lengths exceeding 32K tokens, Star Attention begins to demonstrate clear performance advantages. As demonstrated in Table 1, the speedup achieved by Star Attention increases significantly with longer sequence lengths.

| Seq. Length | Time Per Sample (s) | | |
| (K) | Vanilla | Ring | Star |
|---|---|---|---|
| 16 | 7 | 10 | 9 |
| 32 | 10 | 12 | 10 |
| 64 | 18 | 22 | 12 |
| 128 | OOM | 53 | 20 |

Table 6: Time per sample (seconds) for Llama3.1-8B-Instruct model with vanilla (global) inference, ring (global) and star attention, using 8 A100 GPUs. Vanilla autoregressive generation encounters out-of-memory (OOM) at 128K sequence length. It performs best in short context scenarios (i.e. sequences upto 32K tokens) but in long context scenarios, star attention demonstrates significant speedup.

## C.2 HARDWARE FOR INFERENCE SPEED

In table 1, we use A100 GPUs to run the inference speedup tests. Table 7 describes the number of GPUs and the number of parallel workers used to obtain these numbers for Ring Attention vs Star Attention.

| Model Size | Seq. Length | # GPUs | # Workers |
|---|---|---|---|
| 8B | 16K - 128K | 8 | 4 |
| | 256K - 512K | 16 | 8 |
| | 1M | 32 | 16 |
| 70B | 16K - 32K | 8 | 4 |
| | 64K | 16 | 4 |
| | 128K | 32 | 8 |

Table 7: Resources used for the speedup experiments

## C.3 PROMPT TEMPLATES

Prompt structure for base models:

```
1 {context}{query}{answer_prefix}
```

Prompt structure for Llama-3 and Llama-3.1 Instruct models:

```
1 <|begin_of_text|><|start_header_id|>system<|end_header_id|>
2
3 You are a helpful
    assistant.<|eot_id|><|start_header_id|>user<|end_header_id|>
4
5 {context}{query}<|eot_id|><|start_header_id|>assistant<|end_header_id|>
6
7 {answer_prefix}
```

The portion in **blue** is processed during Phase 1 for blockwise context encoding, while the remaining text in **gray** is processed in Phase 2 for query encoding and token generation. The **{context}** and **{query}{answer_prefix}** denote the context and the query portion of the input prompt, respectively. The **{answer_prefix}** is only relevant for the RULER benchmark.

# D    ACCURACY OF STAR ATTENTION

Table 8 shows the exact accuracy scores of star attention vs global attention across the RULER and the BABILong benchmark from Figure 4.

| Model | Seq. length | Block size | RULER (%) Global | Star | Δ | BABILONG (%) Global | Star | Δ |
|---|---|---|---|---|---|---|---|---|
| GradientAI Llama-3-8B -Instruct-262k | 16K | 4K | 88.92 | 89.48 | +0.63% | 43.60 | 43.40 | -0.46% |
| | 32K | 8K | 85.25 | 85.74 | +0.58% | 40.00 | 39.40 | -1.50% |
| | 64K | 16K | 83.17 | 82.30 | -1.05% | 40.40 | 39.00 | -3.47% |
| | 128K | 32K | 79.25 | 77.79 | -1.83% | 30.80 | 33.20 | +7.79% |
| Meta Llama-3.1-8B -Instruct | 16K | 4K | 99.78 | 91.27 | -1.02% | 59.60 | 59.80 | +0.34% |
| | 32K | 8K | 99.66 | 88.70 | +1.34% | 54.60 | 54.00 | -1.10% |
| | 64K | 16K | 98.72 | 83.37 | -1.67% | 49.20 | 46.60 | -5.28% |
| | 128K | 32K | 92.54 | 74.41 | -2.49% | 40.00 | 38.60 | -3.50% |
| Meta Llama-3.1-8B -Base | 16K | 4K | 77.18 | 78.64 | +1.9% | 22.00 | 25.20 | +14.55% |
| | 32K | 8K | 74.76 | 76.91 | +2.88% | 22.60 | 24.00 | +6.19% |
| | 64K | 16K | 70.01 | 69.09 | -1.32% | 26.80 | 27.20 | +1.49% |
| | 128K | 32K | 64.68 | 69.58 | +7.58% | 31.00 | 26.40 | -14.84% |

Table 8: Accuracy (%) of star attention on RULER and BABILONG evaluated on sequence lengths of 16K, 32K, 64K, and 128K. In all experiments, the block size and anchor block size are set to one-quarter of the total sequence length. Results using the Llama-3-8B-Instruct-262k, Llama-3.1-8B-Instruct and Llama-3.1-8B-Base models demonstrate that star attention retains 95-100% of the accuracy of global attention, and in some cases, even outperform it.

# E    ACCURACY ON ALL RULER TASKS

This section contains the accuracy of all the models we evaluated across all 13 tasks in RULER.

| **Llama-3.1-8B-Instruct** | | | | | | | | | |
|---|---|---|---|---|---|---|---|---|---|
| Block Size (K) | Seq. Len. (K) | Retrieval (NIAH) | | | | | | | |
| | | Single 1 | Single 2 | Single 3 | Multi-Key 1 | Multi-Key 2 | Multi-Key 3 | Multi-Value | Multi-Query |
| Global Attn. | 16 | 100 | 100 | 100 | 99.8 | 100 | 99 | 99.9 | 99.5 |
| | 32 | 100 | 100 | 100 | 99.8 | 99.8 | 99.6 | 99 | 99.05 |
| | 64 | 100 | 100 | 100 | 99.4 | 99.2 | 96.8 | 95.15 | 99.2 |
| | 128 | 100 | 99.6 | 99.8 | 97.6 | 87.2 | 66.8 | 91.55 | 97.8 |
| 4 | 16 | 100 | 99.4 | 100 | 98 | 98.8 | 99 | 91.1 | 98.25 |
| 8 | 32 | 100 | 100 | 100 | 99.2 | 99.4 | 98.2 | 94 | 98.3 |
| 16 | 64 | 100 | 100 | 100 | 99.2 | 98 | 90 | 85.35 | 97.9 |
| 32 | 128 | 100 | 100 | 99.6 | 96.4 | 84.8 | 59 | 82.7 | 96.55 |

Table 9: Accuracy of Llama-3.1-8B-Instruct on retrieval tasks in RULER with Global Attention and Star Attention

| Llama-3.1-8B-Instruct | | Multi-Hop | Aggregation | | Question Answering | |
|---|---|---|---|---|---|---|
| Block Size (K) | Seq. Len. (K) | VT | CWE | FWE | QA 1 | QA 2 |
| Global Attn. | 16 | 99.56 | 75 | 88.87 | 80.8 | 56.4 |
| | 32 | 99.2 | 14.7 | 93.93 | 78.8 | 54 |
| | 64 | 95.44 | 1.96 | 85.13 | 78.8 | 51.2 |
| | 128 | 61.76 | 0.04 | 72.33 | 76 | 41.6 |
| 4 | 16 | 91.96 | 85.72 | 89.73 | 80.2 | 54.4 |
| 8 | 32 | 92.68 | 45.66 | 95.27 | 78.6 | 51.8 |
| 16 | 64 | 92.32 | 5.78 | 86.47 | 78.4 | 50.4 |
| 32 | 128 | 62.8 | 0.04 | 75.87 | 68 | 41.6 |

Table 10: Accuracy of Llama-3.1-8B-Instruct on multi-hop, aggregation, and question answering tasks in RULER with Global Attention and Star Attention

| Llama-3.1-8B-Base | | Retrieval (NIAH) | | | | | | | |
|---|---|---|---|---|---|---|---|---|---|
| Block Size (K) | Seq. Len. (K) | Single 1 | Single 2 | Single 3 | Multi-Key 1 | Multi-Key 2 | Multi-Key 3 | Multi-Value | Multi-Query |
| Global Attn. | 16 | 100 | 100 | 100 | 99.2 | 100 | 99.4 | 99.45 | 99.85 |
| | 32 | 100 | 100 | 100 | 99 | 99.4 | 99.4 | 99.55 | 99.4 |
| | 64 | 100 | 100 | 100 | 98.8 | 86.2 | 95.4 | 96.8 | 97.55 |
| | 128 | 100 | 100 | 98 | 93.8 | 53.6 | 64 | 80.9 | 85.3 |
| 4 | 16 | 100 | 100 | 100 | 97.4 | 99.2 | 99 | 98.4 | 99.15 |
| 8 | 32 | 100 | 100 | 100 | 96.2 | 98.2 | 99.2 | 98.55 | 98.7 |
| 16 | 64 | 100 | 100 | 100 | 96.6 | 90.6 | 85.6 | 94.9 | 96.15 |
| 32 | 128 | 100 | 100 | 98.2 | 88.8 | 67 | 47.6 | 72.75 | 77.55 |

Table 11: Accuracy of Llama-3.1-8B-Base on retrieval tasks in RULER with Global Attention and Star Attention

| Llama-3.1-8B-Base | | Multi-Hop | Aggregation | | Question Answering | |
|---|---|---|---|---|---|---|
| Block Size (K) | Seq. Len. (K) | VT | CWE | FWE | QA 1 | QA 2 |
| Global Attn. | 16 | 99.92 | 65.66 | 17.4 | 11 | 11.4 |
| | 32 | 99.28 | 23.56 | 28.27 | 13.8 | 10.2 |
| | 64 | 96.8 | 2.04 | 13.73 | 14.2 | 8.6 |
| | 128 | 71.68 | 0.64 | 30.53 | 51.2 | 11.2 |
| 4 | 16 | 97.24 | 86.46 | 20.67 | 11.6 | 13.2 |
| 8 | 32 | 97.2 | 58.72 | 30.47 | 11.8 | 10.8 |
| 16 | 64 | 94.44 | 8.86 | 11.2 | 10.6 | 9.2 |
| 32 | 128 | 81.6 | 2.98 | 81.27 | 48.2 | 38.6 |

Table 12: Accuracy of Llama-3.1-8B-Base on multi-hop, aggregation, and question answering tasks in RULER with Global Attention and Star Attention

| Llama-3.1-70B-Instruct | | | | | | | | | |
|---|---|---|---|---|---|---|---|---|---|
| Block Size (K) | Seq. Len. (K) | Retrieval (NIAH) | | | | | | | |
| | | Single 1 | Single 2 | Single 3 | Multi-Key 1 | Multi-Key 2 | Multi-Key 3 | Multi-Value | Multi-Query |
| Global Attn. | 16 | 100 | 100 | 100 | 97.8 | 99.8 | 98.6 | 99 | 99.65 |
| | 32 | 100 | 100 | 100 | 99.6 | 99 | 99 | 99.1 | 100 |
| | 64 | 100 | 100 | 100 | 99.8 | 96 | 97.6 | 95.65 | 99.95 |
| | 128 | 97.2 | 99.2 | 99.4 | 93 | 26 | 27.4 | 92.05 | 92.45 |
| 4 | 16 | 100 | 100 | 100 | 97.4 | 99.4 | 99.2 | 80.9 | 97.2 |
| 8 | 32 | 100 | 100 | 100 | 98.2 | 96.4 | 95 | 87.85 | 97.4 |
| 16 | 64 | 100 | 100 | 100 | 98 | 93.2 | 95.4 | 86.25 | 96.45 |
| 32 | 128 | 98.2 | 100 | 99.4 | 80 | 19.2 | 16.4 | 61.65 | 70.8 |

Table 13: Accuracy of Llama-3.1-70B-Instruct on retrieval tasks in RULER with Global Attention and Star Attention

| Llama-3.1-70B-Instruct | | | | | | |
|---|---|---|---|---|---|---|
| Block Size (K) | Seq. Len. (K) | Multi-Hop | Aggregation | | Question Answering | |
| | | VT | CWE | FWE | QA 1 | QA 2 |
| Global Attn. | 16 | 100 | 99.3 | 97 | 82.6 | 62.4 |
| | 32 | 100 | 94.22 | 98.87 | 80.4 | 59.8 |
| | 64 | 100 | 39.7 | 93.73 | 74.6 | 54 |
| | 128 | 50.08 | 2.98 | 77 | 58.4 | 33.6 |
| 4 | 16 | 87.32 | 99.52 | 97.13 | 82.2 | 60.6 |
| 8 | 32 | 90.08 | 94.5 | 99.2 | 80.2 | 58 |
| 16 | 64 | 91.52 | 49.54 | 94.93 | 73.4 | 53.6 |
| 32 | 128 | 41.4 | 2.7 | 80.07 | 50.8 | 31 |

Table 14: Accuracy of Llama-3.1-70B-Instruct on multi-hop, aggregation, and question answering tasks in RULER with Global Attention and Star Attention

| GradientAI Llama-3-8B-Instruct-262K | | | | | | | | | |
|---|---|---|---|---|---|---|---|---|---|
| Block Size (K) | Seq. Len. (K) | Retrieval (NIAH) | | | | | | | |
| | | Single 1 | Single 2 | Single 3 | Multi-Key 1 | Multi-Key 2 | Multi-Key 3 | Multi-Value | Multi-Query |
| Global Attn. | 16 | 100 | 100 | 99.8 | 99.6 | 100 | 96 | 95.35 | 99.85 |
| | 32 | 100 | 100 | 100 | 99.8 | 100 | 95 | 96.2 | 99.75 |
| | 64 | 100 | 100 | 100 | 98.4 | 99.4 | 91.4 | 97.75 | 99.6 |
| | 128 | 100 | 97.8 | 98.8 | 98.8 | 99.8 | 79.8 | 94.65 | 98.05 |
| | 256 | 100 | 100 | 99.4 | 96.4 | 89.6 | 25.6 | 87.3 | 93.2 |
| 4 | 16 | 100 | 98.4 | 96.6 | 99.6 | 99.4 | 97 | 89.2 | 99.75 |
| 8 | 32 | 100 | 100 | 100 | 99.2 | 99.6 | 96 | 91.6 | 99.7 |
| 16 | 64 | 100 | 100 | 100 | 99.4 | 99.4 | 90 | 91.45 | 99.3 |
| 32 | 128 | 100 | 100 | 100 | 98.4 | 97.8 | 66.8 | 89.3 | 96.8 |
| 32 | 256 | 100 | 99.6 | 98.4 | 91.4 | 53 | 23 | 75 | 81.05 |

Table 15: Accuracy of GradientAI Llama-3-8B-Instruct-262K on retrieval tasks in RULER with Global Attention and Star Attention

| GradientAI Llama-3-8B-Instruct-262K | | | | | | |
|---|---|---|---|---|---|---|
| Block Size (K) | Seq. Len. (K) | Multi-Hop | Aggregation | | Question Answering | |
| | | VT | CWE | FWE | QA 1 | QA 2 |
| Global Attn. | 16 | 95.36 | 42.1 | 91.07 | 80.2 | 56.6 |
| | 32 | 93.88 | 4.5 | 90.53 | 74 | 54.6 |
| | 64 | 92.28 | 0.22 | 82.73 | 69.8 | 49.6 |
| | 128 | 77.88 | 0.36 | 73.27 | 65.6 | 45.4 |
| | 256 | 52.8 | 1.8 | 77.93 | 67 | 37 |
| 4 | 16 | 90.64 | 67.32 | 90.53 | 77.2 | 57.6 |
| 8 | 32 | 92.16 | 19.2 | 89.6 | 73.8 | 53.8 |
| 16 | 64 | 88.6 | 0.4 | 84.13 | 69.6 | 47.6 |
| 32 | 128 | 81.12 | 0.3 | 75.4 | 61.6 | 43.8 |
| 32 | 256 | 72.64 | 1.8 | 81.6 | 61.6 | 33 |

Table 16: Accuracy of GradientAI Llama-3-8B-Instruct-262K on multi-hop, aggregation, and question answering tasks in RULER with Global Attention and Star Attention

| GradientAI Llama-3-8B-Instruct-1048K | | | | | | | | | |
|---|---|---|---|---|---|---|---|---|---|
| Block Size (K) | Seq. Len. (K) | Retrieval (NIAH) | | | | | | | |
| | | Single 1 | Single 2 | Single 3 | Multi-Key 1 | Multi-Key 2 | Multi-Key 3 | Multi-Value | Multi-Query |
| Global Attn. | 16 | 100 | 99.2 | 100 | 99 | 99.6 | 90.2 | 96.1 | 99.25 |
| | 32 | 100 | 100 | 100 | 99.4 | 99.2 | 69.8 | 96.3 | 98.45 |
| | 64 | 100 | 100 | 100 | 99 | 99 | 51.4 | 96 | 98.75 |
| | 128 | 100 | 98.2 | 99.8 | 99.8 | 98.8 | 42.8 | 98.2 | 97.75 |
| | 256 | 100 | 100 | 100 | 98.4 | 97 | 22.4 | 96.1 | 97.15 |
| | 512 | 100 | 99.8 | 100 | 95.6 | 88.4 | 9.4 | 89.25 | 92.55 |
| | 1024 | 99.4 | 99.4 | 100 | 92.6 | 67.8 | 1.4 | 82 | 88.85 |
| 4 | 16 | 100 | 98 | 96.8 | 98.6 | 99 | 94.4 | 90.3 | 98.1 |
| 8 | 32 | 100 | 99.8 | 100 | 98.8 | 99.4 | 87 | 91 | 97.6 |
| 16 | 64 | 100 | 100 | 100 | 99.4 | 99 | 66.8 | 92.3 | 97.95 |
| 32 | 128 | 100 | 100 | 100 | 99.4 | 98.4 | 62.8 | 92.25 | 96.8 |
| 32 | 256 | 100 | 99.8 | 100 | 95.4 | 90.4 | 53.8 | 76.5 | 88.6 |
| 32 | 512 | 99.8 | 95.8 | 97.6 | 85.8 | 64.2 | 19.4 | 57.2 | 63.8 |
| 32 | 1024 | 99.6 | 97.2 | 100 | 84.2 | 27 | 1 | 55.15 | 60.3 |

Table 17: Accuracy of GradientAI Llama-3-8B-Instruct-1048K on retrieval tasks in RULER with Global Attention and Star Attention

| GradientAI Llama-3-8B-Instruct-1048K | | | | | | |
|---|---|---|---|---|---|---|
| Block Size (K) | Seq. Len. (K) | Multi-Hop | Aggregation | | Question Answering | |
| | | VT | CWE | FWE | QA 1 | QA 2 |
| Global Attn. | 16 | 93.32 | 22.52 | 88.73 | 76.8 | 54.8 |
| | 32 | 91.96 | 0.54 | 87.27 | 75.4 | 54.4 |
| | 64 | 81.6 | 0.32 | 79.8 | 75.4 | 46.4 |
| | 128 | 76.68 | 0.22 | 76.67 | 68.4 | 48.8 |
| | 256 | 63 | 0.22 | 78.27 | 71.4 | 43.8 |
| | 512 | 34.8 | 0.86 | 85.6 | 66.8 | 37.8 |
| | 1024 | 24.28 | 3.5 | 72.53 | 66.2 | 30.2 |
| 4 | 16 | 90 | 62.64 | 86.53 | 77.4 | 55.4 |
| 8 | 32 | 90.08 | 11.38 | 85.2 | 75.2 | 53.8 |
| 16 | 64 | 81.96 | 0.38 | 82 | 74.4 | 46.6 |
| 32 | 128 | 79.32 | 0.22 | 78.33 | 64.6 | 46.4 |
| 32 | 256 | 66.4 | 0.22 | 81.53 | 67.4 | 37.6 |
| 32 | 512 | 50.76 | 0.24 | 85 | 63 | 30.8 |
| 32 | 1024 | 63.68 | 6.9 | 79.67 | 57.8 | 26.4 |

Table 18: Accuracy of GradientAI Llama-3-8B-Instruct-1048K on multi-hop, aggregation, and question answering tasks in RULER with Global Attention and Star Attention

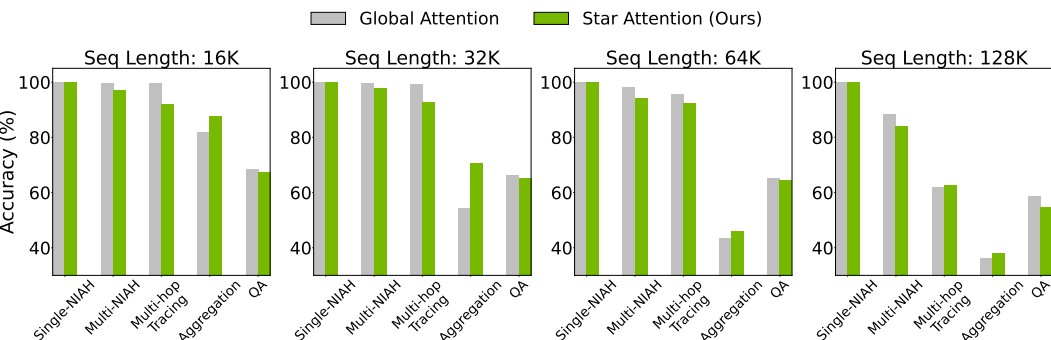

Figure 8: Accuracy (%) of star attention using the Llama-3.1-8B-Instruct model on the 5 categories of tasks in RULER on sequence lengths of 16K, 32K, 64K, and 128K. In all experiments, the block size and anchor block size are set to one-quarter of the total sequence length. For the NIAH and QA tasks, Star Attention retains upto 95-100% accuracy of the baseline. The Multi-Hop Tracing task is notably challenging because it requires inter-block communication, which leads to expected performance degradation. Interestingly, Star Attention performs better with sequence lengths of 128k on this task, but this may be due to noise given the suboptimal baseline. In aggregation tasks, Star Attention show significant improvement as distributed local attention helps the model in such summarization tasks.

