# OpenReview forum: "Star Attention: Efficient LLM Inference over Long Sequences"
_ICLR.cc/2025/Conference — Submitted to ICLR 2025_

### Official Review · Reviewer_dPwG · 2024-10-29

**Soundness:** 3
**Presentation:** 3
**Contribution:** 3
**Rating:** 6
**Confidence:** 3

**Summary:**

The paper introduces Star Attention, an approach designed to enhance the efficiency of large language models (LLMs) in processing extremely long input sequences. This method mitigates the high computational costs associated with traditional global self-attention by optimizing the prefill stage of inference: it employs a local attention mechanism with "anchor" blocks to capture core context across segmented portions of the input. Experimental results show that Star Attention improves inference efficiency while maintaining accuracy levels similar to those of full attention. Further, the paper conducts comprehensive evaluations to validate the rationale and underlying principles of the proposed method.

**Strengths:**

1. This paper provides an in-depth analysis of the "attention sink", presenting a new perspective on it and exploring the underlying causes of this effect.
2. This paper presents a concise and effective approach to accelerate long-sequence processing in LLMs, improving operational efficiency with limited accuracy degradation.
3. This paper provides ablation studies to validate the rationale and potential principles underlying the proposed method.
4. This paper is well-written and well-organized.

**Weaknesses:**

1.  The evaluation section of this paper includes only a few baseline methods. Aside from the global attention used in Ring Attention, no comparisons on accuracy and efficiency were made with other relevant parallel global context computation or attention approximation methods, such as Flash Attention, sliding window attention, or sparse attention.
2.  In Table 2, for the Gradient-AI Llama3-8B-Instruct-1024K at sequence lengths of 512K and 1024K, and the Meta Llama-3.1-70B-Instruct at a sequence length of 128K, an accuracy drop of over 10% can be observed. The paper lacks a more detailed experimental analysis of these scenarios.

**Questions:**

1. I suggest the authors considered compare Star Attention's accuracy and efficiency with other related works. Such comparisons could provide a more comprehensive understanding of Star Attention’s performance relative to existing methods.
2. In cases with longer sequences or larger models, accuracy degradation becomes more pronounced. Have you tried any approaches to mitigate this issue, such as increasing the block size directly? Or is it acceptable?

---

> ### Author Response · Authors · 2024-11-14
>
> We thank reviewer dPwG for their thoughtful comments.
>
> * Thank you for pointing it out. Star Attention is complementary and orthogonal to most of parallel global context computation or attention approximation methods and we have emphasized it in the updated version of the paper.
>   * Taking Flash Attention as an example, in Star Attention, each host processes it assigned context block using Flash Attention.
>   * Star Attention is a inference-only method that does not require any additional training/finetuning. Whereas the methods like Sliding Window Attention etc. require the model to be specifically trained. This is why we don't consider these as a baseline.
>   * As for some of the other sparse attention methods that do not require training, star Attention is orthogonal to all those methods. We can further optimize Star Attention by incorporating these methods in all the hosts. This is why we consider only Ring Attention as the baseline.
>
> * For Star Attention, we show in the paper that as long as the block size is 1/4th of the sequence length, star attention is closely able to match the accuracy of global attention while giving significant speedup. But if reduce the block size to be 1/8, 1/16, etc. of sequence length, the speedup would be even more significant at the cost of even more accuracy degradation.
>   * It was our mistake that we combined both these things in a single table leading to confusion among the readers. We have updated the paper by splitting the table 2 into two parts where in the first part we keep block size ~ 1/4 of seq length. This is our main table.
>   * Then in the second part, we show the impact on accuracy and speedup if we freeze the block size and keep on increasing the seq length.
>   * We also convert the original table 1 to a plot to clearly demonstrate star attention is not applicable to only one benchmark or model but can be used on other models as well.
>
>
> We also wanted to let you know that based on the feedback from other reviewers, we have made some updates to the paper:
> * We updated the Table 1, Table 2 and Figure 3 in Section 3 to now contain the full RULER benchmark. This includes 13 tasks spanning domains such as Retrieval, Multi-Hop Tracing, Aggregation, and Question Answering. Despite including much more complex and diverse tasks, our overall conclusion remains the same.
> * We also added a new subsection (3.4) discussing the particular domain of tasks where Star Attention shows significant advantages over global attention and also the most challenging domains for such two-phase inference approach.
> * In the appendix, we have also listed all the RULER scores for each individual task and each model.
> * We rewrote the text in section 3.3 to make the tradeoff section more clear.
> * We also updated the introduction section slightly to make it clear that that star attention is orthogonal to other training-free sparse attention methods and can be combined with them.
>
> Other changes:
> * We also made the text in Section 2 (that describes Star Attention) much more clearer and we explain it using an example and with simplified equations.
> * We added the star attention pseudo-code and the accuracy on each ruler task in the appendix.

---

### Official Review · Reviewer_JR9S · 2024-11-03

**Soundness:** 2
**Presentation:** 3
**Contribution:** 2
**Rating:** 6
**Confidence:** 4

**Summary:**

This paper proposes star attention, which is based on an assumption "global attention is not necessary during context processing". It breaks the long context into multiple segments, prepend the first block with all other blocks, doing local attention. Then it replicates the queries and compute attention between query and each block. Finally, it sums up all block-wise attention to generate outputs. The method is evaluated with one synthetic benchmark and compare with only one baseline Ring Attention.

**Strengths:**

1. The proposed method is easy to understand.

2. The long-text LLM is an important research problem nowadays.

**Weaknesses:**

1. Lack of baselines and related work. Transformer for long context is an active area. But this paper does not have a related work section to discuss many related work in this area. Can authors justify why the baseline only includes Ring Attention, any many other related methods (e.g., StreamingLLM, Longformer, Reformer, Unlimiformer, are not included in the baselines or discussed in the paper? The papers i list here might already be out-dated, but there definitely are a lot more papers in the last two years.

2. Only one synthetic benchmark is used in the experiment. I think Ruler is a good benchmark, but it is fully synthetic. The proposed method is based on that "global attention is not necessary in context processing", so I think it is important to evaluate the proposed method on real-world data (e.g., InfiniteBench and Bamboo).

3. The accuracy reported in Table 1 and Table 2 seem to be contradictory. In Table 1, the conclusion is the proposed method can maintain 95-100% accuracy. However in Table 2, the delta could be more than 10%. What is the delta in Table 2? If delta is the relative error, then ithe accuracy of Table 2 seems to be a lot worse than the accuracy in Table 1. (e.g., 95 * 90% =85.5).

4. The proposed method cannot be applied to cases where the LLM needs to generate a lot of  tokens (e.g., writing a paper). Though it is not a serious issue, it limits the application and contribution of the proposed method.

**Questions:**

See weakness above.

---

> ### Author Response · Authors · 2024-11-14
>
> We thank reviewer JR9S for their thoughtful comments.
>
> * **Why Ring Attention for Baseline:** In the paper, we show that for these long context inference scenarios, the model does not need to attend to all the previous tokens. This is why the baseline for star attention is full global attention.
>   * But using standard global attention implementation gives OOM for very long sequence lengths. We use ring attention just as a way to implement exact attention in order to be able to run inference on such long sequences with exact attention.
>
> * **Comparison with other techniques:** Star Attention is a inference-only method that does not require any additional training/finetuning. Whereas the methods like StreamingLLM, Longformer etc. require the model to be specifically trained. This is why we don't consider these as a baseline.
>   * As for some of the other sparse attention methods that do not require training, Star Attention is complementary and orthogonal to all those methods. We can further optimize Star Attention by incorporating these methods in all the hosts. This is why we consider only Ring Attention as the baseline.
>   * In the introduction section we already referenced a quite a few of the works and it was a significant overlap to the related work section. Thus we decided to merge those two into the introduction instead.
>
>
> * **Other Benchmarks:** Apart from RULER, we also show the scores on the BABILong benchmark. The babilong benchmark contain several supporting facts throughout the context and the model is required to follow through all the facts in order to be able to answer the question.
>   * We have updated the experiements section in the paper to have the full RULER scores instead of just the NIAH tasks. The full RULER benchmark has 13 tasks spanning domains such as Retrieval, Multi-Hop Tracing, Aggregation, and Question Answering. The Question Answering domain contains two QA tasks that come by combining paragraphs from non-synthetic datasets such as SQuAD and hotpotqa.
>
> * For Star Attention, we show in the paper that as long as the block size is 1/4th of the sequence length, star attention is closely able to match the accuracy of global attention while giving significant speedup. But if reduce the block size to be 1/8, 1/16, etc. of sequence length, the speedup would be even more significant at the cost of even more accuracy degradation.
>   * It was our mistake that we combined both these things in a single table leading to confusion among the readers. We have updated the paper by splitting the table 2 into two parts where in the first part we keep block size ~ 1/4 of seq length. This is our main table.
>   * Then in the second part, we show the impact on accuracy and speedup if we freeze the block size and keep on increasing the seq length.
>   * We also convert the original table 1 to a plot to clearly demonstrate star attention is not applicable to only one benchmark or model but can be used on other models as well.
>   * The delta show in paper is the relative delta i.e. delta = (acc_star - acc_global) / acc_global. Thanks for pointing it out, we have added this clarification in the captions.
>
> * Yes, this is the assumption behind star attention that the length of the context is much greater than the length of the query and output. This conforms to the most common long context inference scenarios where the input context is very long and the model takes a lot of time just to process that long context. Star Attention aims to optimize this context encoding process without degrading the accuracy.
>
>
> We also wanted to let you know that based on the feedback from other reviewers, we have made some updates to the paper:
> * We updated the Table 1, Table 2 and Figure 3 in Section 3 to now contain the full RULER benchmark. Despite including much more complex and diverse tasks, our overall conclusion remains the same.
> * We also added a new subsection (3.4) discussing the particular domain of tasks where Star Attention shows significant advantages over global attention and also the most challenging domains for such two-phase inference approach.
> * In the appendix, we have also listed all the RULER scores for each individual task and each model.
> * We rewrote the text in section 3.3 to make the tradeoff section more clear.
> * We also updated the introduction section slightly to make it clear that that star attention is orthogonal to other training-free sparse attention methods and can be combined with them.
>
> Other changes:
> * We also made the text in Section 2 (that describes Star Attention) much more clearer and we explain it using an example and with simplified equations.
> * We added the star attention pseudo-code and the accuracy on each ruler task in the appendix.

---

> > ### Comment · Reviewer_JR9S · 2024-11-21
> > **Question about benchmark tasks**
> >
> > I appreciate that the authors address most of my concerns. I do have a remaining question about the benchmark task. Is there any task where the output requires summarizing the information from the whole context (e.g., meeting summarization task from QMSum). I ask this question because it makes sense that the proposed method can work when the output only needs a piece of information from a segment of the context. But in meeting summarization or summarizing a whole book, I feel the global attention becomes necessary during context processing. Do authors have comments or results on this?

---

> > > ### Author Response · Authors · 2024-11-21
> > >
> > > Thank you so much for your response. When we evaluate star attention on RULER, it consists of 13 tasks in which **two of those tasks belong to the aggregation/summarization domain: *Common Words Extraction* and *Frequent Words Extraction*.** These tasks require the model to look over the entire context and output the top 10 most common words in the context. In **section 3.4** of the paper, we show that these **summarization/aggregation tasks are actually the domains where star attention shows the most significant improvement.** It clearly outperforms full global attention in terms of accuracy.
> > > This enhancement stems from Star Attention’s chunk-based context division, enabling local attention within each chunk to strengthen summarization capabilities. Effective chunk-wise summarization in Phase 1 facilitates global attention’s information aggregation in Phase 2.
> > >
> > > Apart from summarization, RULER also has a task called Variable Tracking which requires the model to do **multi-hop tracing** throughout the entire context. We see that Star Attention performs well in those tasks as well.
> > >
> > > We provide the individual accuracies on all these tasks in Appendix E.
> > >
> > > As for the currently available summarization tasks such as QMSum, GovReport, etc. The average length of a sequence within those tasks is less than 14K tokens, thus they are not long context.

---

> > > > ### Comment · Reviewer_JR9S · 2024-11-21
> > > >
> > > > Thank you for the clarification. I think the multi-hop tracing task is what I meant.
> > > >
> > > > Originally, my major concern of the paper is the lack of the baseline. But I think the explanation in the rebuttal about why some baselines are not included is reasonable. So I will increase the rating to 6 unless other people can provide some other baselines that this paper should compare with.

---

> > > > > ### Author Response · Authors · 2024-11-22
> > > > >
> > > > > Thank you for taking the time to re-evaluate our paper and for your thoughtful feedback—it’s greatly appreciated!

---

### Official Review · Reviewer_LaPe · 2024-11-03

**Soundness:** 2
**Presentation:** 2
**Contribution:** 2
**Rating:** 5
**Confidence:** 4

**Summary:**

The paper presents Star Attention, a two stage approach to reduce the inference time of long context Transformers. It achieves 11x speed up with 95%-100% the performance.

**Strengths:**

1. The paper method is simple and clear.
2. The experiments are well motivated and provide decent ablations.

**Weaknesses:**

The main experiment is conducted mainly to RingAttention, an exact attention mechanism. In the reviewer opinion, the  experiments should also cover the following dimensions:

(1) Other sparse attention methods, e.g. H2o.
(2) exact attention system: there are other systems than RingAttention that is faster, e.g. DistFlashAttn.

**Questions:**

Please address the weakness section. Thanks!

---

> ### Author Response · Authors · 2024-11-14
>
> We thank reviewer LaPe for their thoughtful comments on our submission.
>
> * Star Attention is orthogonal to all these sparse attention methods. We can further optimize Star Attention by incorporating these KV cache compression methods in all the hosts.
> * In the paper, we show that for these long context inference scenarios, the model does not need to attend to all the previous tokens. This is why the baseline for star attention is full global attention.
>   * But using standard global attention implementation gives OOM for very long sequence lengths. We use ring attention just as a way to implement exact attention in order to be able to run inference on such long sequences with exact attention.
> * Similarly, we can also implement exact attention with DistFlashAttention and then implement star attention using it so that the relative speedup is comparable.
>
> We also wanted to let you know that based on the feedback from other reviewers, we have made some updates to the paper:
> * We updated the Table 1, Table 2 and Figure 3 in Section 3 to now contain the full RULER benchmark. This includes 13 tasks spanning domains such as Retrieval, Multi-Hop Tracing, Aggregation, and Question Answering. Despite including much more complex and diverse tasks, our overall conclusion remains the same.
> * We also added a new subsection (3.4) discussing the particular domain of tasks where Star Attention shows significant advantages over global attention and also the most challenging domains for such two-phase inference approach.
> * In the appendix, we have also listed all the RULER scores for each individual task and each model.
> * We rewrote the text in section 3.3 to make the tradeoff section more clear.
> * We also updated the introduction section slightly to make it clear that that star attention is orthogonal to other training-free sparse attention methods and can be combined with them.
>
> Other changes:
> * We also made the text in Section 2 (that describes Star Attention) much more clearer and we explain it using an example and with simplified equations.
> * We added the star attention pseudo-code and the accuracy on each ruler task in the appendix.

---

### Official Review · Reviewer_iuNN · 2024-11-04

**Soundness:** 2
**Presentation:** 3
**Contribution:** 2
**Rating:** 5
**Confidence:** 4

**Summary:**

This paper introduces star attention, a two-phase inference techniques to handle long sequences efficiently.

The author find that pure blockwise context encoding can bring attention spike in the attention scores , and propose "anchor blocks" to address this issue in stage 1, by attach the first block (the "anchor block") to the begining of each sub-block and conduct the encoding, which shift the attention spike to the "anchor block", and only use the encoded sub-block part for inference. In stage 2, the author combine the online softmax technique to make full use of query and previous encoded sequence for token generation.

The author conduct experiment on long-context benchmarks like RULER-NIAH using Llama family models, and do the abliation study on anchor blocks, learn effect of its position, content and size.

**Strengths:**

- This paper propose a interesting solution "anchor block" for breaking the dependencey of long-sequence to enable process in parallel.
- The ablation study is sufficient and present details effect of "anchor block" from different dimision.

**Weaknesses:**

- Given the technical depth of the proposed method, the experiment result is not strong enough. In Table 1,  the improvement that the star attention can bring is marginal. While In Table 2, the propsoed method is hard able to achieve significant speed up without sacrifiing a huge amount of performance, especially on super long sequence.
- The experimental result may not solid. Combine Table1 and Table2, I notice that Llama-3.1-70B shows clear poor performence comparing to Llama-3.1-8B on both base and instruct version. This observation makes me questionable on the experiment set up and impelmentation.
- There presentation is not complete. For example, Figure 3 left is missing the data point on 32K lenght block, which make it inconsistent to the Figure 3 right. This may due to the seq-lenght limitation of 64K, the author should switch the setting from 64K to 256K to make the figure full-fill. also,Figure 3 type error "instruct global att".

**Questions:**

Follwing the discussion of the weakness part, I have following questions:
1. Why llama3-8B perform much better than llama3-70B on RULER-NIAH in both base and instruct version?
2. How to measure the "speed up"? it is unclear about the which time is used for comparsion, the time to generate the first token or the time to finish the response.
3. Why the star attention clearly outperform the global one in BABLONG and clearly underperform in RULER-NIAH at the same time?

---

> ### Author Response · Authors · 2024-11-14
>
> We thank reviewer iuNN for their thoughtful comments and we also thank you for pointing out the issues with the figures, we have fixed it.
>
> * For Star Attention, we show in the paper that as long as the block size is 1/4th of the sequence length, star attention is closely able to match the accuracy of global attention while giving significant speedup. But if reduce the block size to be 1/8, 1/16, etc. of sequence length, the speedup would be even more significant at the cost of even more accuracy degradation.
>   * It was our mistake that we combined both these things in a single table leading to confusion among the readers. We have updated the paper by splitting the table 2 into two parts where in the first part we keep block size ~ 1/4 of seq length. This is our main table.
>   * Then in the second part, we show the impact on accuracy and speedup if we freeze the block size and keep on increasing the seq length.
>   * We also convert the original table 1 to a plot to clearly demonstrate star attention is not applicable to only one benchmark or model but can be used on other models as well.
>
> * **BABILong Discrepancy**: In Table 1, we see accuracy on RULER and BABILong follow the same trends for the instruct models and this discrepancy only happens with the base model where BABILong improves while the RULER scores drop. As mentioned in the second paragraph of Section 3.2, this fluctuation is because BABILong requires the model to generate answers in a specific format, which is challenging for base model because they are not trained for instruction following. Thus making the scores of the base model on this benchmark unreliable.
>
> * **Llama-8B scoring higher than Llama-70B**: Our results have the same trend as reported in the original RULER benchmark leaderboard: https://github.com/NVIDIA/RULER
>   * On sequences upto 64K, 70B has higher score as compared to 8B. But on 128K, the 8B scores higher (77 for 8B vs 66 for 70B).
>   * Even for the gradient-AI 8B model, its score on 128K sequence length is higher (69.5 for 8B vs 66 for 70B).
>
> * **Inference Time**: When measuring this for a given sequence length, the model is evaluated on a NIAH task from RULER on 500 samples. We start counting the time after the model and the data is loaded and stop when inference is finished. The total inference time in then divided by 500 to get time per sample.
>
>
> We also wanted to let you know that based on the feedback from other reviewers, we have made some updates to the paper:
> * We updated the tables and figures in Section 3 to now contain the full RULER benchmark. This includes 13 tasks spanning domains such as Retrieval, Multi-Hop Tracing, Aggregation, and Question Answering. Despite including much more complex and diverse tasks, our overall conclusion remains the same.
> * We also added a new subsection (3.4) discussing the particular domain of tasks where Star Attention shows significant advantages over global attention and also the most challenging domains for such two-phase inference approach.
> * In the appendix, we have also listed all the RULER scores for each individual task and each model.
> * We rewrote the text in section 3.3 to make the tradeoff section more clear.
> * We also updated the introduction section slightly to make it clear that that star attention is orthogonal to other training-free sparse attention methods and can be combined with them.
>
> Other changes:
> * We also made the text in Section 2 (that describes Star Attention) much more clearer and we explain it using an example and with simplified equations.
> * We added the star attention pseudo-code and the accuracy on each ruler task in the appendix.

---

> > ### Comment · Reviewer_iuNN · 2024-11-29
> >
> > Thank you for your detailed answer. After carefully reviewing the revised version, I decide to maintain my recommendation for weak rejection based on two key concerns with the empirical results:
> >
> > (1)  The proposed method shows no significant improvements beyond the LLaMA3-8B-1024K model in terms of task score, with particularly underwhelming results on the 70B model.
> >
> > (2) The inference framework is based on huggingface, which is typically 10x slower than commonly used one like [vllm](https://github.com/vllm-project/vllm). While I understand the challenges of adding custom features to optimized inference frameworks, the lack of open-source code combined with the use of a slower inference framework makes it difficult to verify 1) the reliability of the reported speed gains comparing to the baseline ring-attn, as the inference speed of ring-attn can heavily depends on the implementation. 2) whether the reported speed gains would translate to real-world implementations.

---

> > > ### Author Response · Authors · 2024-11-29
> > >
> > > Thank you so much for your response.
> > >
> > > 1. The goal of star attention is not to improve upon the baseline method but to give speedup gains while maintaining equivalent accuracy as one would get with doing full global attention.
> > >     * Doing global attention over such long sequences is extremely slow and compute intensive.
> > >     * With star attention, we show a way to use only local attention for a majority part of the context to give significant speedup while maintaining similar accuracy.
> > >
> > > This why from our results, we show that star attention is able to retain upto 95% accuracy of global attention and giving upto 11x speedup. And this is without requiring any additional finetuning.
> > >
> > > 2. Yes, we understand that huggingface is slower as compared to vllm and this is why we show only relative improvements in speedup. Since both star attention and ring attention are implemented using the same framework, their relative speedup difference would remain the same even if we move on to a different framework.
> > >     * We actually did try implementing star attention on the trt-llm framework and saw similar speedup gains there as well.

---

### Meta-Review · Area_Chair_vN1D · 2024-12-20

**Metareview:**

The paper proposes Star Attention, a two-phase approach for efficient long-sequence LLM inference that combines local attention within blocks combined with global attention of final tokens to each block, aiming to reduce computational costs while maintaining model performance.

Unfortunately several concerns remained after author feedback. The main issues were the lack of comprehensive comparisons with other sparse attention methods beyond Ring Attention, and questions about whether the speedup gains would translate to real-world implementations.

**Additional Comments On Reviewer Discussion:**

The author feedback phase was useful as acknowledged by most reviewers. Some of the concerns however remained if the work is ready for the high bar of ICLR.

---

### Decision · Program_Chairs · 2025-01-22

Reject